# Epidemiological Features of Postpartum Subclinical Ketosis in Dairy Herds in Hokkaido, Japan

**DOI:** 10.3390/ani14010144

**Published:** 2023-12-31

**Authors:** Kyoko Chisato, Takerou Yamazaki, Shuji Kayasaki, Rika Fukumori, Shin Oikawa

**Affiliations:** 1Veterinary Herd Health, Department of Veterinary Medicine, School of Veterinary Medicine, Rakuno Gakuen University, 582 Bunkyodai-Midorimachi, Ebetsu, Hokkaido 069-8501, Japan; k-chisato@rakuno.ac.jp (K.C.); fukumori@rakuno.ac.jp (R.F.); 2Memanbetsu Livestock Clinic, Ozora Central Branch, Okhotsk Regional Center, Hokkaido Agricultural Mutual Aid Association, 149-10 Memanbetsu Syouwa, Ozora-cho, Abashiri-gun, Hokkaido 099-2356, Japan; takerou_yamazaki_gb@nosai-do.or.jp; 3Teshikaga Livestock Clinic, Kushiro Central Branch, East Regional Center, Hokkaido Agricultural Mutual Aid Association, 3-10-13 Sakuraoka, Teshikaga-cho, Kawakami-gun, Hokkaido 088-3213, Japan; s22041006@stu.rakuno.ac.jp

**Keywords:** dairy cow, subclinical ketosis, prevalence, risk factor, postpartum

## Abstract

**Simple Summary:**

The aim of this study was to investigate the prevalence of subclinical ketosis (SCK) in Hokkaido, Japan, and to assess its characteristics epidemiologically at the individual and herd levels. Blood samples were taken from clinically healthy cows once within 3–88 days in milk (DIM) for blood tests. Cows with SCK were classified as SCK II within 2 weeks postpartum and SCK I from 15 DIM. The prevalence of SCK II (20.2%) tended to be higher than that of SCK I (16.5%, *p* = 0.094). The prevalence peaked around 2 weeks postpartum. The frequency of SCK I was higher at the fourth parity. The number of milking cows in herds with higher SCK (≥25%) was significantly smaller than in herds with lower SCK (*p* = 0.004). The frequency of herds with higher SCK in tie stalls with component feeding was higher than for those in free stalls and free barns fed a total mixed ration (*p* = 0.054 and *p* = 0.002). This study reveals the prevalence of SCK in Hokkaido, Japan, and shows that SCK is associated with parity and the management system.

**Abstract:**

This study was carried out as an observational study in order to determine the prevalence of postpartum subclinical ketosis (SCK) in dairy herds in Hokkaido, Japan. From April 2012 to March 2014, blood β-hydroxybutyrate (BHBA) concentration was measured once within 3–88 days in milk (DIM) in 1394 apparently healthy cows from 108 farms to diagnose SCK (≥1.2 mM). In cows within 14 DIM, this was classified as SCK II, and from 15 DIM, this was classified as SCK I. Herds with a combined percentage of SCK I and SCK II of less than 10% were classified as SCK-negative herds, those with percentages of 10–25%, were classified as alert herds, and those with one of 25% or more, we classified as positive herds. The prevalence of SCK in the entire DIM was 17.6%. The prevalence of SCK II (20.2%) tended to occur more frequently than SCK I (16.5%, *p* = 0.094). The frequency of SCK I was higher at the fourth parity. The number of milking cows in SCK-positive herds was significantly smaller than those of the other two types of herds (*p* = 0.004). The frequency of SCK-positive herds in tie stalls and with component feeding was higher than for free stall or free barn and with total mixed ration (*p* = 0.054 and *p* = 0.002). This study reveals the prevalence of SCK in Hokkaido, Japan, and shows that SCK is associated with parity and the management system.

## 1. Introduction

The three weeks before and after calving are called the “transition period” [1,2], during which dramatic changes occur in the cows. This means that the fetus grows significantly in the three weeks before calving, and after calving, more energy is required for milk synthesis. At this time, if the energy supply from feed intake is insufficient, cows can easily fall into a negative energy balance (NEB) [3,4]. The cow’s body attempts to adapt to the NEB by utilizing the carbohydrates, lipids, and proteins stored in the body. However, if the reaction is not sufficiently successful, it is assumed that the likelihood of peripartum diseases such as milk fever, retained placenta, and displaced abomasum increases.

Ketosis is known as a fundamental pathology closely linked to peripartum diseases [5]. It is divided into clinical and subclinical ketosis (SCK). A subclinical state is a condition in which ketone body levels are elevated without clinical symptoms. The gold standard for SCK is serum or plasma β-hydroxybutyrate (BHBA) concentration of 1.2 mM or more [6,7]. Furthermore, SCK is classified into two types depending on the time of onset. Type I usually occurs 3 to 6 weeks postpartum (SCK I), and type II occurs within 2 weeks postpartum (SCK II) during early lactation [8]. They were named because of their similarity in pathogenesis to human diabetes mellitus, but recent studies have indicated that their pathogenesis is not consistent, and Mann and McArt (2023) discourage the use of the terminology of type I and type II ketosis [9]. On the other hand, SCK II, which occurs in the early postpartum period, is known to be the underlying disease for many perinatal diseases and does not have a favorable prognosis. But, SCK I is known to have a good response to treatments. We thought that grasping the prevalence of SCK I and SCK II would be useful information for understanding herd health in this study.

Previous studies have shown that the prevalence of SCK varies by country and time of occurrence. Briefly, in the United States, the prevalence within 3–16 days in milk (DIM) was calculated to be 43.2% (range from 26 to 56%), and the peak prevalence of SCK occurred at 5 DIM [10]. A study in 10 European countries reported that the prevalence within 2–15 DIM was 21.8% (range from 11.2 to 36.6%) [11]. In Slovakia, the prevalence of SCK within 20 DIM was calculated to be 18.5%, whereas it was 14.1% for 21–90 DIM [12]. However, there are no reports investigating the prevalence of SCK in Japan.

Several individual factors are known to be associated with the occurrence of SCK, including the body condition score (BCS), parity, dystocia, twins, and calving season [11,13]. Walsh (2007) reported that the prevalence of SCK was 29.8% in tie stalls (TS) and 15.4% in two-row free stalls (FS) [14]. Furthermore, Garzón-Audor and Oliver-Espinosa (2019) estimated that farms with 150 or more cows were at higher risk of SCK occurrence than farms with fewer cows (relative risk: 1.92; 95%CI: 1.04–3.57) [15]. These findings suggest that the management system may be related to the incidence of SCK. Japanese dairy cows are kept in a wide variety of systems, ranging from TS of small herd size to FS and free barns (FB) of medium and large herd size. However, the relationship between the occurrence of SCK and management systems such as housing and feeding is unclear.

The aim of this study was to determine the prevalence of postpartum SCK in Hokkaido, Japan, and evaluate its characteristics at individual and herd levels through epidemiological surveys.

## 2. Materials and Methods

### 2.1. Animals and Their Information

This epidemiological study was conducted as an observational study between October 2012 and March 2014 in 18 regions of Hokkaido, Japan, through regular health examinations of blood profiles and physical condition by 11 veterinarians. The 108 farms (size range: 23–800 cows) that were insured by the Hokkaido Agricultural Mutual Aid Association and agreed to participate in this study were selected as survey farms. A total of 1407 clinically healthy Holstein cows, free of clinical disease for at least one week prior to blood sampling, were studied. Each veterinarian performed an average of 2.1 visits per farm (SE: 0.1, range: 1–10). The average number of sampling cows per farm was 12.9 (SE: 0.8, range: 1–63). Records for farm name, individual identification number, date of calving, and parity were obtained from the database of the Livestock Mutual Aid System of the Agricultural Mutual Aid Association. This accurately reflects the traceability information of cows as required by domestic law. Dairy farm information such as housing system (TS and FS/FB), feeding system (total mixed ration (TMR) and component feeding), and feeding frequency was also obtained through oral investigation conducted by veterinarian at the time of blood sampling.

Blood BHBA concentration ≥ 1.2 mmol/L was diagnosed as SCK II within 3–14 DIM and as SCK I from 15 DIM. On the other hand, cows with BHBA < 1.2 mM were classified as healthy controls. This study investigated the relationships between parity (1, 2, 3, 4, and 5 or more), BCS (≤3.25 or 3.50≤), RFS (≤2 or 3≤), housing and feeding systems, and number of cows with SCK.

Farms used for herd analysis were those with 12 or more cows sampled per farm [7]. Herds with a combined percentage of SCK I and SCK II of less than 10% were classified as SCK-negative herds, those with percentages between 10 and 25% were classified as SCK-alert herds, and those with one of 25% or more were classified as SCK-positive herds [7]. This study investigated the relationship between SCK-negative, -alert, and -positive herds and management systems (average number of cows, housing system, feeding system, and feeding frequency). Table 1 shows the collection method, type of variable, and definition used in individual and herd analyses.

All animals were treated appropriately following the Laboratory Animal Control Guidelines of Rakuno Gakuen University, which essentially conforms to the Guide for the Care and Use of Laboratory Animals of the National Institutes of Health in the United States [16].

### 2.2. Blood Sampling and Physical Examinations

Blood samples were collected from the coccygeal vein only once per cow during 3–90 DIM. Plain collection tubes were used to collect serum for measurement of the concentrations of BHBA and nonesterified fatty acids (NEFA). Collection tubes with sodium fluoride were used for measuring the glucose concentration. Collected blood was stored immediately at 4 °C. For serum separation, centrifugation was performed at 2000× *g* for 15 min within 6 h after collection, and sera were stored at −30 °C until analysis. The concentrations of serum BHBA, NEFA, and glucose were measured using an automatic analyzer (Bio Majesty JCA-BM2250; JEOL, Tokyo, Japan) with kits for them (N-assay NEFA; Nittobo Medical, Tokyo, Japan, 3-OHBA-TR; Kishimoto Clinical Laboratory, Sapporo, Japan, and GLU-TR; Kishimoto Clinical Laboratory) from the Kishimoto Clinical Laboratory. Cows were scored for BCS [17] and the rumen fill score (RFS) [18] together with blood collection by same veterinarian per farm. The BCS was rated on a 5-point scale with increments of 0.25, with higher scores indicating higher body fat stores. Plus, RFS was also assessed on a 5-point scale in increments of 1, with higher scores indicating better dry matter intake. All veterinarians used the same scoring system.

### 2.3. Statistical Analysis

All statistical analyses were performed using SPSS version 27.0 (IBM Japan Co. Ltd., Tokyo, Japan). Continuous data were transformed as necessary. A log_10_ transformation was applied to BHBA and NEFA. Other outcomes did not require transformation. Normality of data was assessed using the Shapiro–Wilk test. The comparison between groups in parametric data was analyzed by ANOVA and Games–Howell test. The analysis of non-parametric data was performed by Kruskal–Wallis test.

The chi-square test was used to compare categorical data (BCS, RFS, parity, housing system, feeding system, and feeding frequency). When significance was calculated to be present by the chi-square test, adjusted residuals below −1.96 were rated as significantly low frequency of occurrence and above +1.96 as significantly high frequency of occurrence [19].

The analysis of blood metabolites in healthy and SCK cows was assessed by a linear mixed model. The statistical model was the following:
*Y_pq_* = *μ* + *Type_p_* + *Herd_q_* + *e_pq_*
where *Y_pq_* is the observed value (BHBA, NEFA, Glucose); μ is the overall mean; *Typep* is the fixed effect of the *p*th class of type (*p* = healthy control, SCK I, SCK II); *Herdq* is the random effect of the *j*th herd (*q* = 1–108); and *epq* is the residual error. Bonferroni’s multiple comparison test was used when comparing with groups.

Statistically significant differences were assessed at *p* < 0.05, and a trend was assessed at *p* < 0.10.

## 3. Results and Discussion

### 3.1. Analysis of Individual Level

#### 3.1.1. Prevalence of SCK and Comparison of SCK I and SCK II by Blood Test

Of the cows sampled, seven and six cows were excluded due to a lack of physical examination data and treatment on the day of sampling, respectively. As a result, data from 1394 cows within 3–88 DIM were used.

The prevalence of SCKI and SCKII classified by DIM are shown in Figure 1. The prevalence of SCK in overall DIM was 17.6% (245/1394). The minimum and maximum prevalence were 10.5% (15/143) at 3–7 DIM and 25.6% (67/262) at 8–14 DIM, respectively. Table 2 shows serum BHBA concentrations classified by DIM. Similar to the prevalence, the serum BHBA concentration was significantly lower (*p* = 0.017) at 3–7 DIM (0.82 ± 0.03) and higher at 8–14 DIM (1.06 ± 0.05) and 22–28 DIM (1.08 ± 0.06). The prevalence of SCK has been reported in different regions of the world. Suthar et al. (2013) reported that the prevalence of SCK at 2–15 DIM was 21.8% (ranging from 11.2% to 36.6%) in European countries [11]. Additionally, Duffield et al. (1997) calculated the prevalence of SCK to be 14.1% within 65 DIM [20]. Therefore, the prevalence of SCK in the present study is similar to those reported in Europe and North America.

Until the 1980s, the peak prevalence of SCK was 3 to 4 weeks postpartum [21,22], but since the late 1990s, it has shifted to within 2 weeks postpartum [11,20]. Moreover, McArt et al. (2012) reported that the peak of SCK prevalence was seen at 5 DIM [10]. It has been suggested that advances in genetics, housing, and feeding management have brought the challenges of energy metabolism closer to calving [20]. The peak SCK prevalence in Hokkaido was also observed around 2 weeks postpartum, which may suggest that energy metabolism has changed in the early postpartum period.

Table 3 shows the result of the comparison of blood metabolites in SCK I and SCK II based on a linear mixed model. The serum NEFA concentration was higher in SCK II than in SCK I and healthy controls, like the BHBA concentration (*p* < 0.001). Cows with SCK I and SCK II had significantly lower glucose concentrations than healthy controls (*p* < 0.001). The changes in metabolites of SCK I and SCK II in the present study were similar to those in previous studies [8,23,24]. The names of SCK I and SCK II were derived from the fact that their pathological conditions are similar to human diabetes, though their onset times and mechanisms are different [8]. In general, SCK I occurs in cows whose milk production is so high that glucose demand exceeds glucose production capacity. Therefore, blood glucose and insulin concentrations are low. On the other hand, it is presumed that insulin and glucose concentrations in SCK II cows are higher initially and then become lower because the pathological state is associated with insulin resistance [23]. However, it has also been reported that no such pathophysiological changes are seen in SCK II through experimental study, raising questions about the merits of the nomenclature [9]. In fact, no hyperglycemia was observed in the present data.

#### 3.1.2. Comparison of BCS, RFS, and Parity in SCK I and SCK II

Table 4 shows the results of statistical analysis (the chi-square test) of the prevalence of SCK categorized by BCS and RFS. The SCK I showed a significantly lower frequency of BCS 3.50≤ and a significantly higher frequency of BCS ≤ 3.25. On the other hand, the results for SCK II were exactly the opposite (*p* = 0.018). In general, BCS reflects feeding status one month prior to monitoring. Garro et al. (2014) suggested that cows with prepartum BCS ≥ 3.75 had a 5.25 (95% CI: 1.32–21.11) times greater risk of developing SCK than cows with prepartum BCS ≤ 3.50 [25]. In this study, the frequency of SCK II cows was significantly higher at 3.50 ≤, suggesting that the BCS may already be higher in the late dry period before calving.

Meanwhile, in SCK II cows, the frequency of RFS 2 or less (unacceptable level) was significantly higher, and the frequency of RFS 3 or higher (acceptable level) was significantly lower (*p* = 0.013, Table 4). Cows with higher BCS at calving reduce dry matter intake during early to mid-lactation [26,27]. Goldhawk et al. (2009) showed that cows diagnosed with SCK within the first week after calving had lower dry matter intake, fewer visits to the feeder, and spent less time at the feeder than healthy animals during the week before calving and the two weeks after calving [28]. The BCS of SCK II cows in this study was significantly higher, which may have caused a decrease in dry matter intake and a decrease in feeding frequency and time, as described above.

Table 5 shows the results of statistical analysis (the chi-square test) of the prevalence of SCK categorized by parity. The frequency of healthy controls was significantly higher for parity 1, but that of SCK I cows was lower. Although for parity 4, healthy controls had a significantly lower frequency, it was higher for SCK I cows (*p* < 0.01). It is known that the prevalence of SCK increases in multiparous cows [20,29]. Suthar et al. (2013) suggested that within 2–15 DIM, cows with a parity of 4 or more were 2.5 times more likely to develop SCK than primiparous cows [11]. Multiparous cows increase milk yield with increasing parity [30]. Increased milk production associated with higher parity may have increased the prevalence of SCK. However, milk production was not investigated in this study, so further investigation is needed in the future.

#### 3.1.3. Association of SCK I and SCK II Prevalence with Management System

The number of farms and cows in terms of housing system was FS/FB with 50 farms and 862 cows (mean ± SE, median, IQR; 248.9 ± 25.1, 180.0, 261.0) and TS with 58 farms and 532 cows (66.7 ± 4.5, 57.0, 25.3). In addition, TMR was used in 75 farms, 1085 cows, and component feeding was used in 33 farms, 309 cows.

Figure 2 shows the association of SCK I and SCK II prevalence with the management system. The prevalence of SCK II, 20.2% (82/405), tended to be higher than that of SCK I, 16.5% (163/989, *p* = 0.094). The prevalence of both SCK II and SCK I was significantly higher (*p* = 0.032 and *p* < 0.001) in TS (26.9% and 21.8%) than in FS/FB (17.5% and 12.7%). This result is consistent with previous studies that suggested the incidence of ketosis was 1.6–2.6 times higher in TS than in FS [31,32].

No relationship between the herd size for each housing type was observed in SCK II cows. However, in TS, the prevalence of SCK I tended to be higher in herds with 57 cows or fewer (*p* = 0.078). The incidence of ketosis was higher in small herds (20 cows) than in medium and large herds (50 cows) in Norway [31]. Furthermore, milk yield was higher at 60–80 DIM and during the late lactation period in TS than in FS [31].

The feeding system in all FS/FB was TMR. However, in the TS system, it was either TMR or component feeding, and the prevalence of SCK I and SCK II were higher with component feeding (*p* = 0.007 and *p* = 0.002). Berge and Vententen (2014) investigated the relationship between the incidence of ketosis and the feeding system in Western Europe and found that the risk of SCK was higher with partial mixed ration (PMR) than with TMR and component feeding [29]. Furthermore, the risk of ketosis was 1.5 times higher in cows on PMR than in those on TMR. The feeding of most herds was with TS, and for those with less than 57 cows, it was component feeding, which may have resulted in inadequate feeding during periods of increased milk production. This study could not identify any factors by which component feeding facilitates an increase in SCK prevalence. Therefore, further investigation is required in the future regarding the feeding system, including PMR.

### 3.2. Analysis of Herd Level

Of the 108 farms, 42 farms with fewer than 12 cows’ blood sampling were excluded from herd-level analysis, and data from 66 farms were ultimately used. One SCK-positive herd was not included in the feeding frequency analysis due to unknown feeding frequency.

The relationships between SCK-negative, -alert, and -positive herds and the housing and feeding systems are shown in Table 6. Of the 66 herds used in the analysis, 16 (24.2%) were classified as SCK-negative herds, 28 (42.5%) as alert herds, and 22 (33.3%) as positive herds (Table 6). It was calculated that 75.8% of the total herds required measures for SCK improvement. This result was similar to that of a previous study which screened 37 herds in the United States and calculated that 78.3% required SCK measures [7].

The SCK-positive herds had significantly fewer milking cows (112.9 ± 26.2) than SCK-negative and -alert herds (241.2 ± 54.4, 207.5 ± 32.1, *p* = 0.004). Berge and Vetenten (2014) reported that large herd size may reduce the risk of developing clinical ketosis because herd composition and feeding content can be adjusted for each lactation stage [29]. However, it is unclear whether the farms used in this study could be grouped by lactation stage, and further investigation is needed.

Additionally, regarding the housing system, SCK-positive herds tended to be more frequently observed in TS (15 farms) than in FS/FB (7 farms, *p* = 0.054). As for the feeding system, the utilization of TMR was higher in SCK-negative herds, and that of component feeding was higher in SCK-positive herds (*p* = 0.002). In terms of feeding frequency, few farms with SCK-negative herds fed more than three times per day, whereas more farms with SCK-positive herds fed more than three times per day, and fewer fed once a day (*p* = 0.083). The more practical feeding frequency in SCK-negative and -alert herds was “twice”, which was different from SCK-positive herds. Although SCK-positive herds were fed more frequently, it was suggested that they were being managed with insufficient energy provision.

## 4. Conclusions

The peak of SCK prevalence was observed at two weeks postpartum. The prevalence of SCK II was higher than that of SCK I. Regarding parity, SCK I had a higher frequency with the fourth parity. At the herd level, SCK-positive herds had significantly fewer cows than the other two groups and higher utilization of TS and component feeding. This study clarifies the prevalence of SCK in Hokkaido, Japan, and shows that SCK is associated with parity and management systems. The obtained results are based on a first-time investigation in Japan and indicate the strength and external validity of this study. However, a limitation is that data might have some “volunteer bias” because of the agreement taken from farmers in performing this study.

## Figures and Tables

**Figure 1 animals-14-00144-f001:**
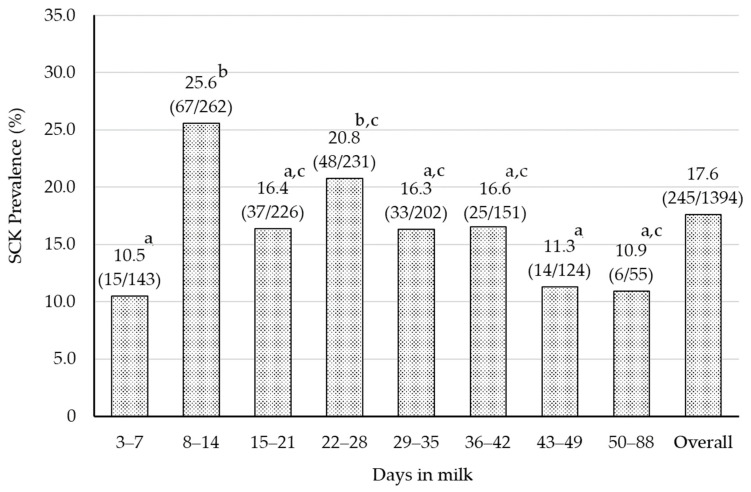
The prevalence of SCK categorized by DIM. The numbers in parentheses in the figure are the number of SCK cows divided by the number of cows tested. The chi-square test was used to determine that there was a significant difference between different superscripts (*p* = 0.001). Days in milk were separated by one-week intervals. Sampling was not conducted during 1–2 DIM due to the influence of calving. Cows in each week from week 8 to and throughout week 13 were treated as the same group because the numbers were limited.

**Figure 2 animals-14-00144-f002:**
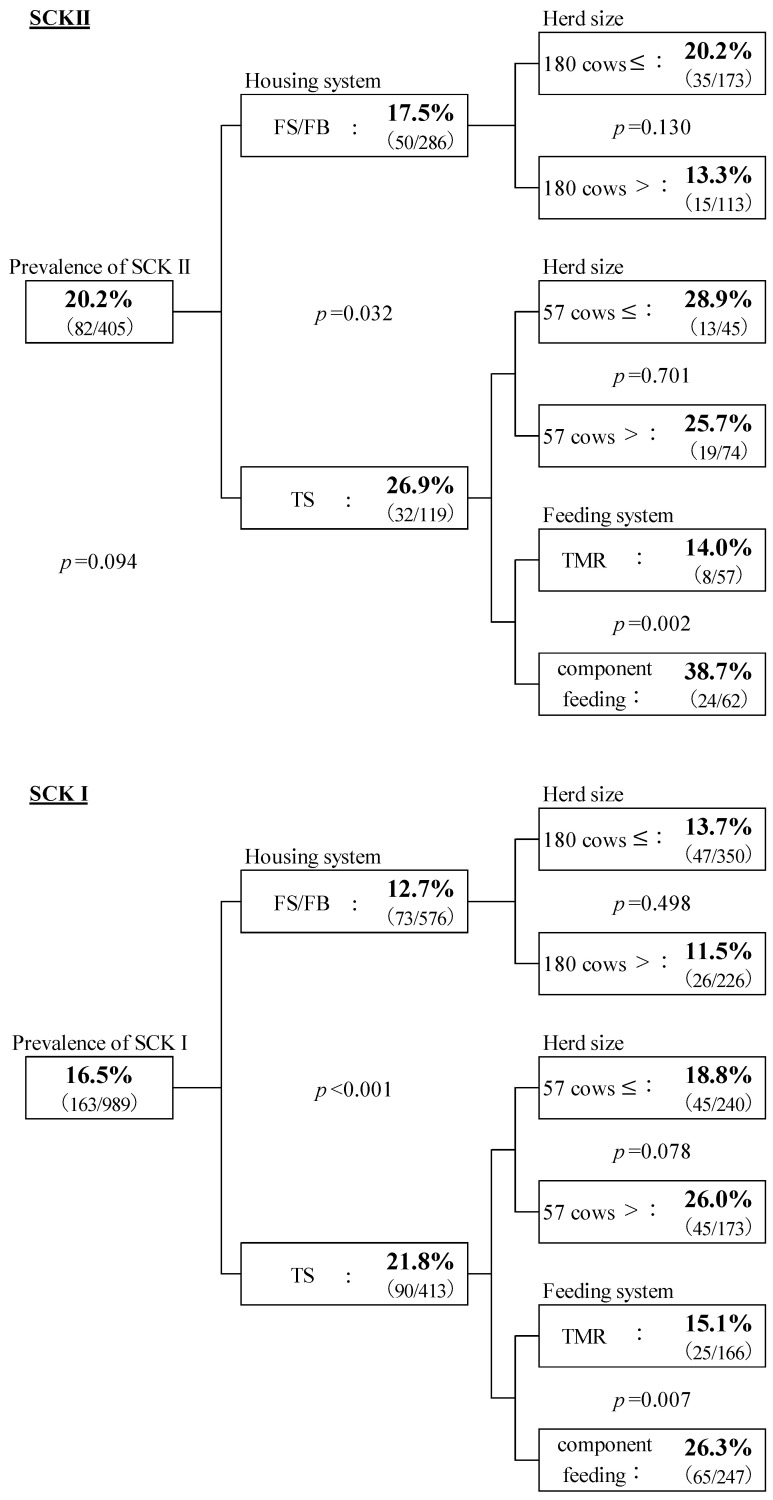
The prevalence of SCK I and SCK II and association with management systems. The chi-square test was used, and statistically significant differences were assessed at *p* < 0.05 and tendency at *p* < 0.10. The percentages in each category show the prevalence of SCK, and the numbers in parentheses show the number of SCK cows and those tested. Housing system: FS/FB = free stall or free barn; TS = tie stall. The average number of milking cows in each housing system was calculated. FS/FB = 180 cows; TS = 57 cows. Farms with more than the median number of cows are considered medium/large-scale farms, and those with fewer than cows are small-scale farms. Feeding system: TMR = total mixed ration; component feeding. Feeding systems of FS/FB are all TMR.

**Table 1 animals-14-00144-t001:** The collection methods, type of variable, and definition used in this study.

Variables	Collection Method	Type of Variable	Definition
BCS	Monitoring	Categorical	≤3.25 or 3.50≤
RFS	Monitoring	Categorical	≤2 or 3≤
Parity	Database	Categorical	1, 2, 3, 4, or 5≤
No. of cows	Database	Categorical/Continuous *	FS/FB: 180> or 180≤
			TS: 57> or 57≤
			/continuous number
Housing system	Oral investigation	Categorical	FS/FB or TS
Feeding system	Oral investigation	Categorical	TMR or component feeding
Feeding frequency	Oral investigation	Categorical	1, 2, or 3≤

* The median number of cows was used as categorical data for comparison of SCK prevalence for small and medium/large herd sizes (see Figure 2). Continuous data were used for SCK herd-level evaluation (negative, alert, and positive, see Table 6).

**Table 2 animals-14-00144-t002:** The serum BHBA concentration categorized by DIM.

DIM ^1^	*n*	Mean ^2,3^	SE ^3^	Median ^3^	IQR ^3^	95%CI ^3^	*p*-Value
3–7	143	0.82 ^a^	0.03	0.74	0.33	0.76–0.89	0.017
8–14	262	1.06 ^b^	0.05	0.82	0.64	0.96–1.15	
15–21	226	0.98 ^a,b^	0.06	0.74	0.45	0.86–1.09	
22–28	231	1.08 ^b^	0.06	0.77	0.50	0.96–1.19	
29–35	202	0.97 ^a,b^	0.06	0.77	0.39	0.86–1.08	
36–42	151	0.90 ^a,b^	0.05	0.70	0.36	0.80–0.99	
43–49	124	0.93 ^a,b^	0.07	0.74	0.34	0.78–1.07	
50–88	55	0.87 ^a,b^	0.04	0.83	0.39	0.78–0.96	

^1^ Classification is the same as in Figure 1. ^2^ Comparisons between groups for each DIM were performed using ANOVA and Games–Howell test. Values showed significant differences between different superscripts (*p* < 0.05). ^3^ A log_10_ transformation was applied for analysis. Obtained data were all back transformed and added to table as final results.

**Table 3 animals-14-00144-t003:** Comparison of blood metabolites in SCK I and SCK II.

	Healthy Control ^1^	SCK I ^2^	SCK II ^3^	*p*-Value
Variable	(*n* = 1149)	(*n* = 163)	(*n* = 82)	
BHBA ^4^ (mM)	0.85 ± 1.01 ^a^	1.35 ± 1.01 ^b^	1.30 ± 1.02 ^b^	<0.001
	(0.84–0.86)	(1.32–1.38)	(1.26–1.36)	
NEFA ^5^ (mEq/L)	0.53 ± 1.02 ^a^	0.63 ± 1.03 ^b^	0.74 ± 1.03 ^c^	<0.001
	(0.52–0.55)	(0.60–0.66)	(0.69–0.78)	
Glu (mg/dL)	56.2 ± 0.44 ^a^	45.9 ± 0.69 ^b^	43.3 ± 0.90 ^c^	<0.001
	(55.3–57.1)	(44.5–47.3)	(41.5–45.0)	

^1–3^ A BHBA concentration of <1.2 mM was considered healthy control, and ≥1.2 mM was diagnosed as SCK [7]. Cows with SCK occurring within 14 DIM were classified as SCK II, and from 15 DIM, they were classified as SCK I [8]. All data were expressed as estimated mean ± standard error. Numbers in parentheses indicate 95% CI. ^4,5^ A log_10_ transformation was applied to BHBA and NEFA for analysis. Obtained data were all back transformed and added to table as final results. The comparison between groups was analyzed by Bonferroni’s multiple comparison test. Values showed significant differences between different superscripts (*p* < 0.05).

**Table 4 animals-14-00144-t004:** Comparison of BCS and RFS in SCK I and SCK II.

		Healthy Control ^1^	SCK I ^2^	SCK II ^3^	*p*-Value
Variables		(*n* = 1149)	(*n* = 163)	(*n* = 82)	
BCS	≤3.25	994	150 ^A^	65 ^a^	0.018
	3.50≤	155	13 ^a^	17 ^A^	
RFS	≤2	281 ^a^	43	32 ^A^	0.013
	3≤	868 ^A^	120	50 ^a^	

^1–3^ A BHBA concentration < 1.2 mM was considered healthy control, and ≥1.2 mM was diagnosed as SCK [7]. Cows with SCK occurring within 14 DIM were classified as SCK II, and from 15 DIM, they were classified as SCK I [8]. The numbers in each variable represent the number of cows. The chi-square test was used, and statistically significant differences were assessed at *p* < 0.05. Capital letters indicate a significantly higher frequency of SCK, and lower-case letters indicate significantly lower one [19].

**Table 5 animals-14-00144-t005:** Comparison of parity in SCK I and SCK II.

		Healthy Control ^1^	SCK I ^2^	SCK II ^3^	*p*-Value
Variable		(*n* = 1149)	(*n* = 163)	(*n* = 82)	
Parity	1	298 ^A^	15 ^a^	16	<0.001
	2	357	55	24	
	3	215	36	18	
	4	131 ^a^	30 ^A^	15	
	5 or more	148	27	9	

^1–3^ A BHBA concentration < 1.2 mM was considered healthy control, and ≥1.2 mM was diagnosed as SCK [7]. Cows with SCK occurring within 14 DIM were classified as SCK II, and from 15 DIM, they were classified as SCK I [8]. The numbers in each variable represent the number of cows. The chi-square test was used, and statistically significant differences were assessed at *p* < 0.05. Capital letters indicate a significantly higher frequency of SCK, and lower-case letters indicate a significantly lower one [19].

**Table 6 animals-14-00144-t006:** Relationships between SCK-negative, -alert, and -positive herds and housing and feeding systems.

Variables	SCK-Negative ^1^ (16 Farms)	SCK-Alert ^1^ (28 Farms)	SCK-Positive ^1^ (22 Farms)	*p*-Value
No. of cows ^2^				
Mean	241.2 ^a^	207.5 ^a^	112.9 ^b^	0.004
SE	54.4	32.1	26.2	
Max	700	800	417	
Min	40	40	23	
Housing system (herds) ^3^				
FS/FB	10	18	7	0.054
TS	6	10	15	
Feeding system (herds) ^3^				
TMR	15 ^A^	22	10 ^a^	0.002
Component feeding	1 ^a^	6	12 ^A^	
Feeding frequency per day (herds) ^3,4^				
Once	3	9	2	0.083
Twice	10	12	8	
Three or more	3	7	11	

^1^ Herds with combined percentages of SCK I and SCK II of less than 10% are classified as negative herds (16 farms), those with percentages of 10–25% are classified as alert herds (28 farms), and those with 25% or more are classified as SCK-positive herds (22 farms) [7]. ^2^ Data are expressed as mean and SE. The Kruskal–Wallis test was used to test significance. There is a significant difference between different letters. ^3^ The chi-square test was used to determine statistical differences at *p* < 0.05 and tendency at *p* < 0.10. Capital letters indicate a significantly higher frequency of SCK, and lower-case letters indicate significantly lower one [19]. ^4^ One SCK-positive herd was not included in the analysis due to unknown feeding frequency.

## Data Availability

Data presented in this study are available upon request from the corresponding author.

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
