# Peer review of "Epidemiological Features of Postpartum Subclinical Ketosis in Dairy Herds in Hokkaido, Japan"

_animals, 2023, doi:10.3390/ani14010144_

Round 1

Reviewer 1 Report

Comments and Suggestions for Authors

The manuscript describes the prevalence of subclinical ketosis in Hokkaido, Japan and shows that this postpartum disorder is associated with parity, the calving season, and the management system. The novelty of the manuscript, as authors were noted, is that they performed for the first time investigations of the prevalence of subclinical ketosis in dairy cows in Japan. I am not agree with the author statment that the prevalence of subclinical ketosis and epidemyological factors are not decsribed in other countries. Many studies investigated the epidemiological features of subclinical ketosis from many years ago but I noted on the attachment file some of the last investigations describes in the review arcticle from 2022. The manuscript contains many errors and mistakes, and most of them I noted in the attachment. I think that will be better if the authors present individually section "Results" and section "Discussion". Especially for section "Discussion", because in the current form it is very informative. I suggest to authors better explanation of their obtained results and to discuss this results with cited references. In conclusion, after making the necessary corrections and changes, the manuscript will be suitable for publication.

Author Response

Thank you very much for taking the time to review this manuscript. We really appreciate your suggestive comments. Please see the attachment.

Reviewer 2 Report

Comments and Suggestions for Authors

Brief summary:

This manuscript reports the prevalence of subclinical ketosis across herds in Hokkaido, Japan. The authors identify associations between subclinical ketosis and cow- and herd-level variables.

General Concept Comments:

Providing observational data on different analytes across different countries and management systems is such an important research topic as it helps provide baseline data and information on how dairy production systems can improve. That being said, I have many concerns with the current manuscript.

First and foremost, I would strongly urge the authors to read the recent publication by Mann and McArt (DOI: 10.1016/j.cvfa.2023.02.004) and reconsider using the terms SCK I and II. How concerned should we be if a cow has a blood BHBA concentration ≥ 1.2 mM beyond 14 DIM? What are the implications of an elevated blood BHBA concentration beyond 14 DIM?

How relevant is calving season for cows with SCK I, particularly given the range in DIM. Wouldn’t it make more sense to evaluate season at the time of blood sample collection? Data has demonstrated that during warm months, BHBA concentrations tend to increase, likely due to decreased DMI. Calving season and current season will mostly be the same for cows evaluated for SCK II, so not an issue there. I would expect a stronger association between season at blood sample collection with SCK than calving season.

Please provide a section discussing strengths, limitations, and external validity of your research.

I highly suggest following STROBE-VET reporting guidelines. These guidelines will help make this a more succinct and influential paper. In reference to those guidelines, here are my specific concerns with the current manuscript:

1)      Abstract: Indicate this was an observational study. How many farms were included?

2)      Setting: How frequently were herds visited? Once?

3)      Participants: What were the eligibility criteria for farms to participate and for cows selected (i.e. were all “apparently healthy looking cows” sampled?) How were farms identified? How many cows per farm were sampled (Stated in ln 127-128, but should be moved earlier in materials and methods when describing general study design)?

4)      Variables: clearly define all outcomes and variables included in models. How were categorical variables created (this is explained for some, but not all variables)? How was BCS assigned (i.e. briefly describing scoring system used in Ferguson paper, how many people assigned scores, same person/people?) Same for RFS.

5)      Data sources/measurement:  Additional information should be included regarding the survey (i.e approximately # of questions, time to complete, type of questions (short answers, multiple choice, etc), who filled it out, how it was administered, and content of questions (i.e. topics of questions, such as questions regarding management, or herd size, etc).

6)      Bias: What efforts were done to prevent selection bias of cows and farms?

7)      Study size: Missing sample size equation or describe how study size was arrived at. Specifically, you have a large range of cows samples per farm (1 to 62). How is 1 cow a representative sample for a farm? This will certainly bias results.

8)      Quantitative variables: This ties in with the Variables section above: How were quantitative variables groupings chosen (such as herd size and DIM)?

9)      Statistical methods: Was farm included as a random effect? If not, it should be. One would argue that cows within a farm are more similar to one another than across farms. In addition, one would argue that BCS and RFS are categorical variables, not continuous, just like parity and feeding frequency. Therefore, a chi-square test would be more appropriate. Additional groupings may be necessary for BCS and RFS and a Fisher’s exact test may be more appropriate if you have no more than 5 cows in a category. In addition, if using an ANOVA, LS means and SE (or if a transformation was done, geometric means and 95% confidence intervals) should be reported in the results section, not mean and SD. Based on your results, multiple comparisons were done. How were P-values corrected for multiple comparisons? What test was used

10)   Participants – In the results section, this is where you should state the number of cows removed due to missing data, number of observations removed from a model due to being an outlier, etc. See my second comment below in the specific comments.

11)   Descriptive data on exposures and potential confounders: I suggest including a paragraph at the beginning of the results section describing (i.e. providing n, mean, SD, median, IQR, range for continuous variables and frequency for categorical variables) the herds (i.e. herd size, freestall vs. tiestall, etc) and the variables of interest (e.g. season). Thoroughly describing these herds would be beneficial to dairy producers and herd advisors in Japan and data may be used for comparison purposes.

12)   Strengths and Limitations: Please provide a section discussing strengths and limitations

13)   Generalizability: What is the external validity of your research?

Specific comments:

Ln. 18: Please do not start a sentence with an abbreviation. This applies throughout the manuscript.

A lot of results are presented in the materials and methods (i.e. ln 89, 95-96, stating # of farms included in analysis (in ln. 102 and 104-106), 108-109, 130-131).

Ln. 152: Do not use the word “tended” if you are not reporting a p-value.

Ln. 173: Just because a cow does not have SCK I or II does not mean she is “healthy.” I suggest changing this to say “cows with BHBA < 1.2 mM.

Ln. 186: change 15 DIM to 14 DIM. SCK I was classified as occurring starting at 15 DIM, not after 15 DIM.

Ln. 224-225: Although I agree that it is likely that multiparous cows had increased prevalence of SCK (particularly SCK I) due to increase milk yield, you cannot make this bold statement as you did not evaluate milk yield. You can not make any inferences as to why the prevalence of SCK was greater in multiparous cows due to 1) not evaluating milk yield and 2) this is an observational study and you cannot make cause and effect claims.

Figure 2: Although I tend to like figures better, I don’t think this is the best way of expressing this data. The outliers have a huge influence on how the figure is presented, making it nearly impossible to detect differences; therefore, the median looks fairly consistent across all DIM due to the scaling of the figure. I suggest providing a table that reports n, mean, SD, median, IQR, and range for each DIM range., and therefore, it is difficult to observe differences in the median concentration across DIM ranges.

Ln. 346-247: significant differences based on what? How was this decided?

Author Response

(The authors gave the same response as above.)

Round 2

Reviewer 1 Report

Comments and Suggestions for Authors

The authors have taken into account and corrected almost all the remarks made in the previous review. In my opinion, the manuscript is suitable for publication in its present form.

Author Response

Thank you very much for taking the time to review this manuscript. We really appreciate your pertinent suggestion.

Reviewer 2 Report

Comments and Suggestions for Authors

Thank you for your responses. You have drastically improved this manuscript through your edits and in particular, the materials and methods section is much clearer. However, I still have serious concerns regarding the statistical analysis and the way some of the results are presented. In addition, some of my previously noted concerns were not addressed.

Specifically, table 3. Based on your statistical analysis explanation, BHBA and NEFA were log transformed. If you are presenting the results from the ANOVA analysis and stating where differences were observed using those log transformed variables, then those variables must be back transformed and the geometric means and 95% CI must be presented from the model results. The superscripts should be applied to the reported geometric means for those log-transformed variables, as the statistical analysis was conducted on the log-transformed variable. You can (and encouraged based on the STROBE-VET guidelines) also present the unadjusted means (which is what you currently have); however, it must be stated that these are the unadjusted means.

In addition, it is not clear what outcomes you are investigating in your statistical analysis section, nor is it clear what your model was. I suggest including the mathematical equation that represents each model.

It was not clear in your response whether herd was included as a random effect in your models. I suggest revising the cow-level analysis (table 3 analyses) and including herd as a random effect, especially due to the vast differences in number of cows sampled per farm. This will help remove some of the variation at the farm-level.

For clarity, can you please include in the manuscript if cows were only sampled once? Or were they allowed to be sampled at subsequent visits? Based on the first manuscript submitted, it seemed like only once; however, that is not clear anymore.

In addition, I have a few comments that I made previously that were not addressed in your response. Please see below:

In the abstract, include that this was an “observational study.” The study design should always be included in the abstract.

You have not included a sample size equation or described how your study size was arrived at. This is hinted at by citing the Borchardt paper (ln. 109); however, this is not enough information. Why did you choose 10 cows as the cut-off for the herd-level analysis? See the sample size section in the Suthar paper as an example.

The groupings chosen for DIM were not consistently 7 days. Why was 50-88 DIM grouped together? A simple explanation, such as “sample size was limited within each week from week 8 through 13 and therefore cows were grouped together,” would be sufficient. Assuming this is the reason. In addition, cows were sampled from 3-7 DIM.

Please do not just tell me what the strengths, limitations, and external validity of this manuscript are. Please include this information in the manuscript. This is important information for the reader. Again, please refer to the STROBE-VET guidelines. https://www.equator-network.org/reporting-guidelines/strobe-vet-statement/

Comments on the Quality of English Language

Minor edits detected. Otherwise, quality is acceptable. 

Author Response

Thank you for your critical comments. Please see the attachment.
